# Heat Treatment of AA7075 by Electropulsing and DC Current Application

**Tyler Grimm ***  **and Laine Mears**

Department of Automotive Engineering , Clemson University, Greenville, SC 29607, USA
* Correspondence: tgrimm@clemson.edu

**Abstract:** Electrical resistivity was used in this test methodology to estimate the relative precipitate density in AA7075. Various electrical test parameters were explored to determine the difference between pulsed and DC-type currents. No difference between these test conditions could be distinguished. Furthermore, an electroplastic effect was not needed to explain these results and the effects are likely to be caused by purely joule heating.

**Keywords:** electroplasticity; retrogression heat treatment; electropulsing; electrically assisted manufacturing



## 1. Introduction

Electrically assisted manufacturing collectively refers to the application of electricity to assist a manufacturing process. More specific to the work discussed herein, this process is used to assist heat treatment processes by passing an electric current through the material. According to some publications, this results not only in resistive heating but other non-thermal effects, known as electroplastic effects.

There exists some literature related to the electropulsing heat treatment of various metals. In general, these authors attempt to make the conclusion that at an equivalent temperature profile, greater effects are observed utilizing resistive heating methods, rather than traditional furnace or other conduction-based heating methods. This work is extensive and this review will only attempt to cover studies that investigated AA7075 or similar alloys, as this is the primary material used herein.

Electrical pulse retrogression (EPR) effects have been cited for several aluminum alloys [1,2]. With these methods, a retrogression heat treatment can be performed in a relatively short time (<30 s). These authors conclude that this is only achievable as a result of an electroplastic mechanism; traditional heat treatment processes cannot induce these changes in such a short amount of time. However, no works to date have attempted a direct comparison between electrical and conduction heating retrogression heat treatments (i.e., the temperature profiles of electropulsed and conventional heated samples were similar). Such a comparison can be attempted with existing literature, though, and is demonstrated in the following text.

A study by Xu et al. [3] investigated electropulsing effects on AA7075-T6. Their testing consisted of applying a pulse of approximately 200 A/mm$^2$ for 220 ms to a sample of AA7075-T6. They observed significant solutionization of precipitates, concluding that this was the result of an electroplastic mechanism since the material never exceeded the solid solution temperature of the secondary phases (≈480 °C).

The authors of [3] analytically determined the maximum temperature of the sample to be 207 °C. However, a numerical simulation was performed to gain better insight into the exact temperature profile of this test. This simulation was performed using COMSOL 6.0. Temperature-dependent material properties were used. The room-temperature values used are listed in Table 1. Note that these references include the temperature-dependent properties that were also used. Convection cooling was applied to all open faces. Since the cooling conditions were not published by Xu et al. [3], various values were tested.

**Table 1.** Material properties of AA7075 used in thermal simulation.

| Specific Heat (J/(kg-K)) | Density (g/cm³) | Thermal Conductivity (W/(m-K)) | Resistivity (μΩ-cm) |
|---|---|---|---|
| 838 [4,5] | 2.72 [6,7] | 124 [4] | 5.86 (Measured value from samples used herein) |

The results of this study are summarized in Figure 1. It is more likely that the specimen reached a maximum temperature of 240 °C and remained at an elevated temperature for about 25 s.

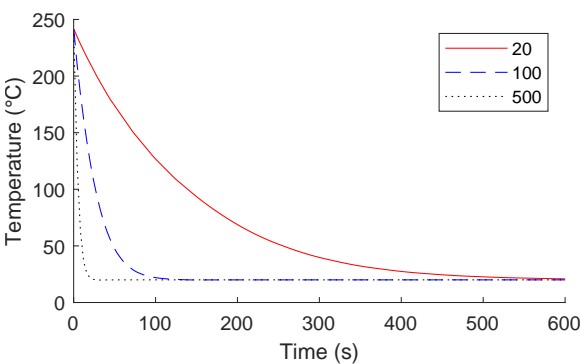

**Figure 1.** Thermal analysis of experiments performed in [3]; Legend values indicate convection coefficient (W/(m²K)).

The results obtained with this temperature profile using resistive-heating methods can now be compared to conventional heating experiments with similar temperature profiles to deduce if an electroplastic mechanism is needed to explain these results. Figure 2 presents the results of a study that used such conventional heat treatment methods (heating performed with immersion into a salt bath or silicon oil bath). Note that this study presented hardness measurements. While hardness was not measured by Xu et al. [3], the resulting tensile strength showed a significant decrease after electropulsing, indicating a reduction in hardness. Figure 2 shows a distinct reduction in hardness after 20–40 s at 220–240 °C. Note that an exact comparison cannot be drawn between these studies since the exact temperature profile of either study was not published. However, this provides a good indication that the results of Xu et al. [3] do not require an electroplastic explanation. These results are simply due to pure thermal heating.

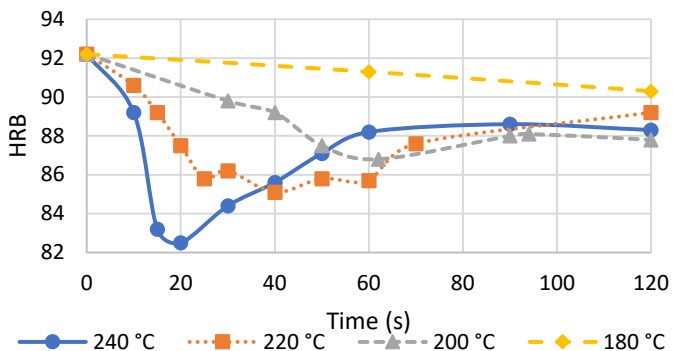

**Figure 2.** Retrogression hardness of AA7075; Reprinted from Publication [8], with permission from Elsevier.

A drawback of the study performed by Xu et al. [3] was that only one set of parameters was explored. The objective of the present study was to gain further understanding of the electropulsing effects on precipitates and, therefore, the resulting strength of aluminum alloys, for a variety of different pulse and cooling conditions. As part of a larger set of work

on this subject, the influence of transient currents is investigated by directly comparing pulsed currents with DC currents.

*Resistance-Based Precipitate Determination*

One method of estimating precipitate density that was used in this research is through the measure of resistivity. This method is not novel and has been performed generally since at least the 1950s. It has been used in several electrically assisted manufacturing works as well [9–11].

There are several drawbacks to this method. The primary goal of this work was to discover if solutionization effects on precipitates can ultimately be used to explain EPEs since precipitate density can generally be linked with strength in aluminum alloys. Unfortunately, the precipitates in AA7075 that contribute mostly to resistance ($Al_6Mn$) are not the same as those which contribute most to strength ($Al_7CuFe$) [12]. Therefore, resistance measurements cannot inherently provide a certain indication of the resulting material strength. This has been overcome in the past by either coupling or substituting resistivity measurements with hardness [13]. Hardness results generally agreed with resistivity results for this material though [14].

## 2. Methodology

To simplify testing, the cross-section of the test sample must be as small as possible. This enables relatively low current amplitudes to produce high current densities. If low current amplitudes can be used, commercial devices such as waveform amplifiers can be used to deliver high-frequency pulses. Sheets of AA7075-T6 of 0.4 mm thickness were sourced for this application.

These sheets were then sheared into thin strips. Shearing was selected as an appropriate method since it will have minimal edge effects relative to other cutting processes. Numerical simulation results influenced the length and width of these strips. The target for the geometry selection was to produce a region of constant temperature which can then be sampled for resistance measurements.

COMSOL 6.0 Multiphysics was used for all numerical simulation work. A 2D model was used, with symmetry applied in both X and Y. To enable high throughput, a fixture was constructed that presses the edges of the specimen between electrodes. The electrodes are much larger than the specimen's cross-section. Therefore, a constant temperature boundary condition was applied on the outer edges of the specimen to simulate the grips/electrodes as heat sinks. Note that this condition should not be applied across all EAM tests, as outlined in a previous publication. This is only applicable when performing rapid pulse testing and the grips are significantly larger than the test specimen [15]. These boundary conditions are shown in Figure 3. Mapped meshing was used with 10 elements in Y and 50 in X. This meshing is shown in Figure 4. Other critical material properties used in this simulation are shown in Table 1. Note that the values listed in this table represent room temperature properties, though the actual simulation utilized temperature-dependent properties from the same references.

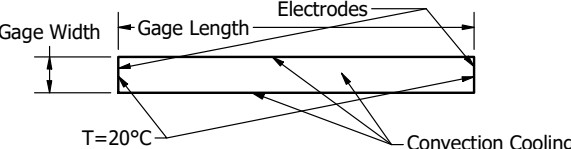

**Figure 3.** Boundary conditions used in numerical simulation. Convection cooling also applied on opposite face.

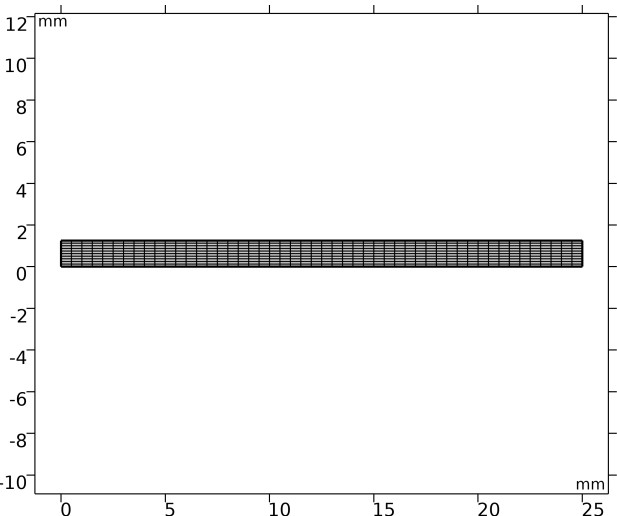

**Figure 4.** Example mesh used for numerical simulation.

Many different parameter combinations were explored. The conclusions of this extensive study are summarized. The results of varying the specimen length are shown in Figure 5. This result used a current density of 250 A/mm$^2$, convection coefficient of 25 W/(m$^2$K), gage width of 2.5 mm, and pulse width of 500 ms. It is clear that a gage length of 100 mm or greater is necessary to produce a moderately sized region of relatively constant temperature. Under these test conditions, a sample region within 5% of the temperature at the center of the specimen (maximum temperature) would be nearly 70 mm long. It was concluded to use a gage length of 100 mm, with a sample region of 75 mm. It was found that the width of the specimen had little effect on the resulting temperature. To reduce the required current amplitude of testing, the width should be made as small as possible, which will depend on the accuracy to which the material can be sheared. Furthermore, the specimen should be small to increase its resistance, according to $R = \frac{\rho l}{A}$. This will enable more accurate resistivity measurements since it will increase the range of measured resistances (e.g., it is easier to measure a 10% difference of 100 than it is of 1). Resistance could also have been increased by increasing the length of the sample. However, a longer sample would also introduce uncertainty in width along its length, as this becomes harder to control the longer the specimen is.

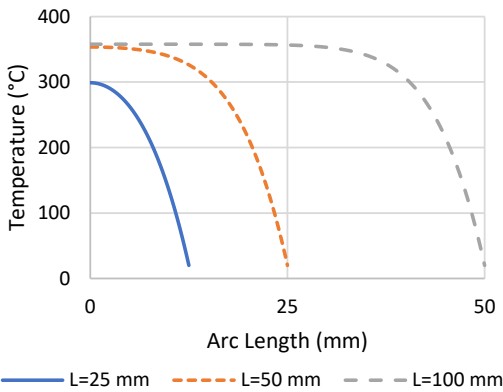

**Figure 5.** Summary result of gage length variation.

As previously stated, the specimen cross-section must be as small as possible. A simple method of cutting sheet specimens is to use a shear. However, it is known that shearing can significantly alter the microstructure of the material along the cut edge. This is usually insignificant when the sheets are large (i.e., the damaged microstructure accounts for a small portion of the total material). However, when cutting very thin strips, this region

may become significant. This should be avoided to maintain repeatability between samples. While it is true that every sample would have similar damage, it should be noted that cutting is a stochastic process. It is better for the majority of the sample to contain the bulk microstructure rather than the damaged microstructure.

To determine the depth of damage from shearing, the microstructure from several cut specimens was analyzed (the baseline microstructure of the -T6 condition is shown in Figure A1 in the Appendix A). An Accucutter 2001 EVO shear was used for all cutting. Several pieces of material were sheared and mounted. The mounted samples were polished to 1 μm and etched with Kellers Reagent for 45 s. Images of the etched samples are shown in Figure 6. The damaged section of these specimens only extends about 50 μm into the specimen at most, with significant damage only extending to about 25 μm. Relative to the thickness of the specimen (400 μm), this damage can be considered insignificant since the majority of the sample will contain the bulk microstructure.

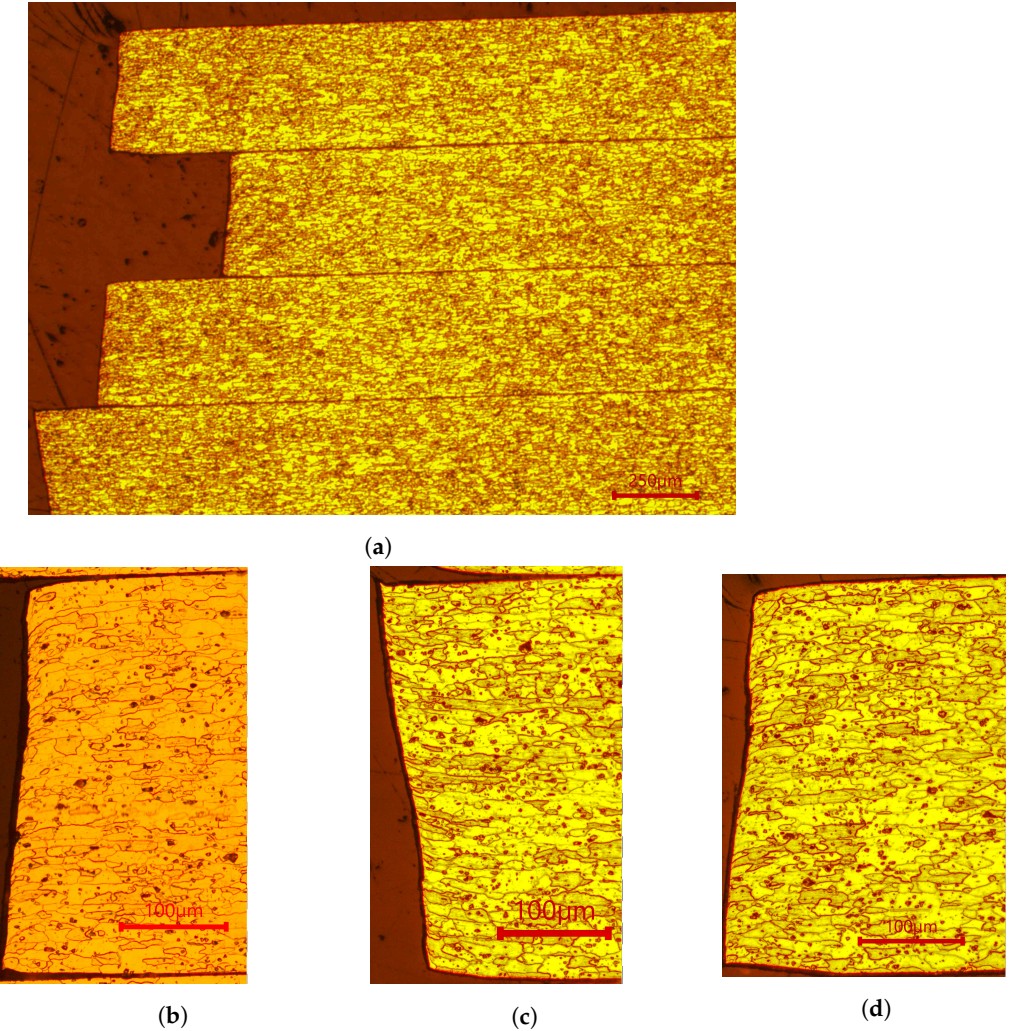

(a)

(b)     (c)     (d)

**Figure 6.** Microstructure of AA7075 sheets after shear cutting; (a) image of several samples, (b–d) close-up images of select samples.

A critical aspect of this study is the repeatability of the specimen size. Since the thickness of the material is fixed, only the width is of concern. A fixture was constructed to aid in repeatably shearing the material to the same width. Many cuts were made and measured using micrometers. A standard deviation of 60.6 μm was determined, which did not significantly change depending on the target width.

Using this standard deviation value, a target width could be determined. If the target width is relatively small, causing the standard deviation to be a significant percentage of

the width, then there would be no way to determine any significant resistance differences. In this case, resistance changes could be due to changes in the specimen width rather than resistivity changes caused by the alteration of precipitates. Additionally, this would cause a significant change in current density between samples since the current amplitude was held constant during testing.

High and low resistivity values of 7075 were derived from [16] to be about 10.6 and 6.5 μΩ-cm. A square cross-section of $0.4 \times 0.4$ mm will be explored as an example. This cross-section will give an area of $0.165 \pm 0.021$ mm$^2$ (±values indicate one standard deviation). With a 50 mm length, the resistance measurements at the maximum resistivity value could be between 28.5–36.8 mΩ, and at the minimum resistivity value could be between 17.5–22.54 mΩ. While a significant difference could still be determined using these values, it would be difficult to interpret any trends in the data due to this uncertainty. When targeting a width of 0.89 mm, these values change to 13.8–15.7 and 8.4–9.6 mΩ, a much more manageable uncertainty. It was determined to target a value close to 0.89 mm (the actual average width will be measured during experimentation).

### 2.1. Baseline Heat Treatment

Several different solutionizing heat treatment cycles for this material were found in the literature, ranging from temperatures of 465–550 °C, times of 20 min–8 h, and quenching in water ranging from room temperature to 65 °C or in room temperature glycol [17–19]. A heat treatment of 500 °C for 2 h followed by a water quench at room temperature was initially used. However, it was discovered that the resulting resistivity of these samples was greater than the -T6 condition, which is the opposite of what is expected from a solutionization heat treatment. Therefore, other heat treatment schedules were tested. Ten repetitions of each condition were tested. These results are shown in Figure 7.

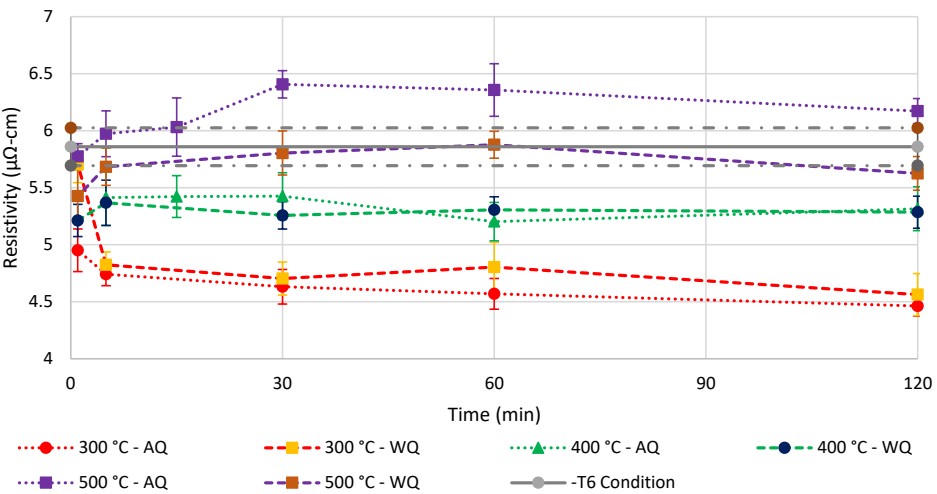

**Figure 7.** Resulting resistivity of different heat treatment cycles; Legend values indicate furnace temperature; AQ—air quenched, WQ—water quenched.

In addition to the water-quenched tests previously discussed, similar testing was performed for the air-quenched condition. The samples were simply removed from the furnace and allowed to cool in room-temperature air for these tests. These data provide some indication of the purely thermal effects on aging and will give insight into the additional effects of electric currents. A limitation of this approach is that furnace heating cannot achieve the same heating rate as the electrically-assisted tests. A numerical simulation was performed to approximate the heating rate from the furnace. The results are shown in Figure 8. This figure also shows the cooling rate in room-temperature air. Temperatures over 500 °C can be reached using electricity in less than one second. However, at the maximum temperature of the furnace used for this testing (1150 °C), it takes about 9.3 s to reach 500 °C. This test

was attempted and produced the results of $5.61 \pm 0.18$ and $5.64 \pm 0.20$ for air-quench and water-quench conditions, respectively. However, this testing was difficult to perform and should only be considered generally. Even with this attempt at rapid conventional heating, the exact conditions of electrically assisted tests could not be replicated for furnace heating.

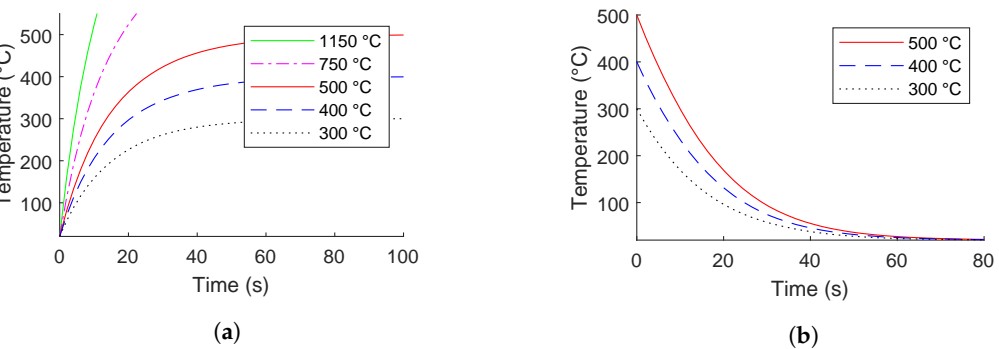

**Figure 8.** Simulation results of furnace heating and cooling; Legend values represent furnace temperature (**a**) or initial temperature (**b**).

This material is well known to age rapidly after solutionization. One study found that after just 15 min at room temperature, the mechanical properties of the material can change [19]. Therefore, samples were immediately stored at $<-30$ °C after quenching (dry ice was used for chilling) or were immediately tested. Samples were removed from this chilled environment and tested within 15 s. Following testing, the samples were allowed to cool to room temperature for 15 s, then placed back into the chilled environment. Samples tested at the -T6 condition were only chilled after testing, since the -T6 condition is not susceptible to aging.

### 2.2. Electrical Application

The fixture shown in Figure 9 was used to apply current to the specimens. The current was applied along the roll direction. This application is generally used in EAM research since bar stock is often used to construct tensile specimens. Since bar stock is generally rolled along the length, tensile specimens cut from these bars will have their axis oriented parallel to the roll direction. This clamping fixture enabled consistent clamping force and distance between the electrodes. These factors ensure consistent resistance within the test circuit between tests, which was critical since voltage-controlled power supplies were used. If the resistance changed between tests, the resulting current would also change.

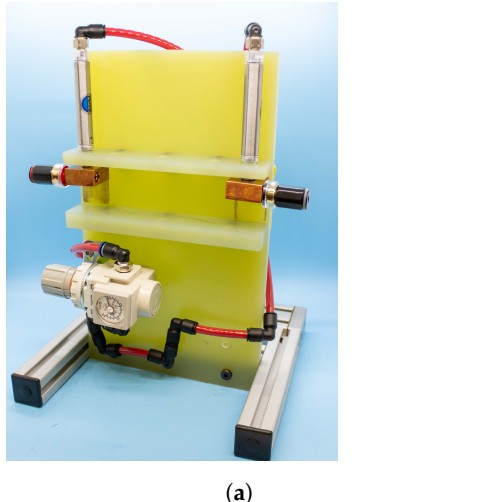
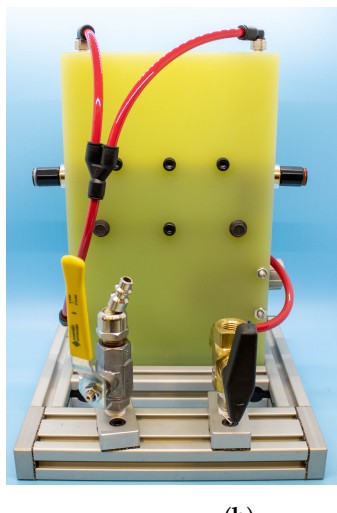

**Figure 9.** Fixture used to apply current to specimens; (**a**) front view, (**b**) back view.

A power supply capable of rapidly pulsing electric current through the specimens was necessary for this work. Target test conditions were derived from the test methodology of Xu et al. [3], who observed solutionization of AA7075 from electropulsing. Their pulse conditions reached a peak current density of 282.8 A/mm$^2$ at an AC frequency of 50 Hz. For the target sample size discussed herein (0.41 × 0.64 mm cross-section; 0.26 mm$^2$), this current density corresponds to a current amplitude of about 75 A. Therefore, the power supply used in this work should at least be capable of reaching this value.

The maximum ramp rate from the testing of Xu et al. [3] is unknown. As a minimum, it can be stated that the power supply must be capable of producing 50 Hz AC frequency. To be clear, the maximum ramp rate from this literature example is likely to be much greater, which would occur when the current is turned on. Ramp rates commonly experienced in DC rectifiers and other solid-state relays are on the order of $10^9$ A/s. While this forms a challenging target value, it is clear that the ramp rate of the power supply should be maximized as much as possible.

An Ametek Sorenson SGA 10/1200 was tested for the present test configuration to confirm its capability. These data are shown in Figure 10. A slow response to the signal generator would be evident by smooth edges or otherwise distorted wave shapes, which is not apparent in this graph (an 18 Hz test is shown in Figure A2 in the Appendix A). Current from this device was measured using a Pico TA018 current probe or a Riedon 696-SSA-1000-ND current probe. Note that both of these current probes can measure frequencies well within the limits of this testing. The waveform of this figure is a 16 Hz triangle wave with a maximum ramp rate of 13.9 kA/mm$^2$/s. This was determined to be the maximum achievable frequency with this device. This frequency/ramp rate was not considered sufficient for this testing. The use of another power supply is discussed in a later section, though this device was still used for low-frequency and DC testing (the use of the Ametek power supply for DC testing is validated in the Section 4, which is presented later).

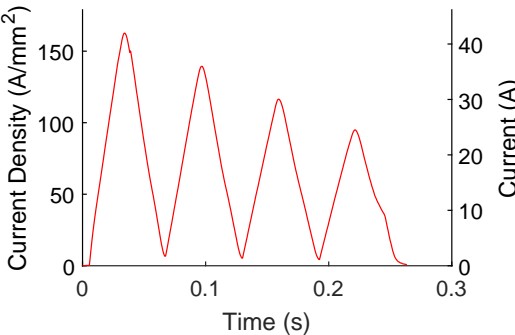

**Figure 10.** Current waveform response of Ametek Sorenson SGA 10/1200 power supply.

In addition to current amplitude, other essential data required for analyses included the instantaneous resistance of the specimen. This information is required to accurately determine the energy input to the sample using the $P = I^2R$ relationship. These data were collected by measuring the voltage drop across the sample, which was accomplished by inserting measurement leads between the power supply leads and the test fixture. Unfortunately, the amplitude of this voltage measurement was heavily skewed, likely from some electrical noise within the circuit or extraneous heating of other elements within the circuit, such as at junction points. This produced unreasonably high resistance values and therefore resulted in extreme temperature rise calculations (>1000 °C). Even after performing a tuning process to eliminate extraneous resistance in the sensing circuit, this result was still observed. Therefore, only the measured current amplitude was used in the calculations discussed herein.

To attempt to distinguish between thermal and non-thermal electrical effects, different cooling and quenching conditions were performed. This included air-cooled, air-quenched, air-cooled, water-quenched, and water-cooled, water-quenched. The air-cooled, water-quenched testing was performed by dropping the specimen into room-temperature water

immediately after electricity was applied. The specimens were quickly released from the test fixture by using an electrically-controlled pneumatic solenoid valve. This valve was triggered immediately after pulsing was complete. The fixture was elevated above the water bath and rotated such that once the clamps were released, the specimen would fall into the bath. The water-cooled, water-quenched tests were performed by placing the fixture inside a CNC machine and using its coolant as the cooling medium. Figure 11 shows this test and displays the aggressiveness of this cooling method. The thermal analyses performed herein were not adjusted for this test condition. For example, one could apply a significantly greater convection coefficient for these experiments which would more closely represent the forced-cooling case. However, since the temperature was not physically measured, a new convection coefficient would introduce a significant assumption. To avoid misrepresenting the data, these values will be noted to be non-physical estimations. These values do, however, represent the relative amount of electric current which has passed through the sample (i.e., a high temperature value can be related to a high amount of electrical energy input).

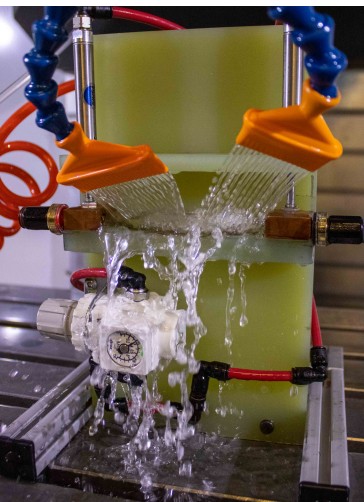

**Figure 11.** Water-cooling, water-quench test condition.

### 2.3. Thermal Estimation

The hypothesis tested in this work, and by EAM research in general, is that varying electrical parameters can cause different physical effects regardless of thermal effects. Therefore, it was desired to determine the thermal difference between tests performed herein.

Several test conditions were selected that approximately span the maximum ranges of testing. These tests are listed in Table 2. The measured current from these tests is shown in Figure 12. These current profiles were used in a similar simulation to what was previously described in this section. As previously mentioned, in situ resistance measurements should be feasible with this test approach when utilizing the Ametek Sorenson SGA 10/1200 power supply, as this enables the use of a CSR in the test circuit. This would have offered much greater accuracy of temperature estimations. Since these data were erroneous and could not be used, the alternative method was to use the baseline resistivity for all calculations, even though the resistivity should increase as the sample's temperature increases. Note that this method is typically used in EAM literature, especially with aluminum alloys where tables of resistivity at elevated temperatures cannot be easily produced due to aging effects. While this does not necessarily represent an acceptable standard, as will be discussed in this section, the resulting uncertainty has been addressed and will be considered in the conclusions made herein.

**Table 2.** Example tests used for thermal analyses.

| Test ID | Frequency (Hz) | Time (s) | $T_{final}$ (°C) |
|---|---|---|---|
| 1 | 2 | 0.5 | 470.5 |
| 2 | 8 | 0.5 | 494.2 |
| 3 | DC | 0.1 | 468.4 |
| 4 | DC | 1.0 | 488.5 |

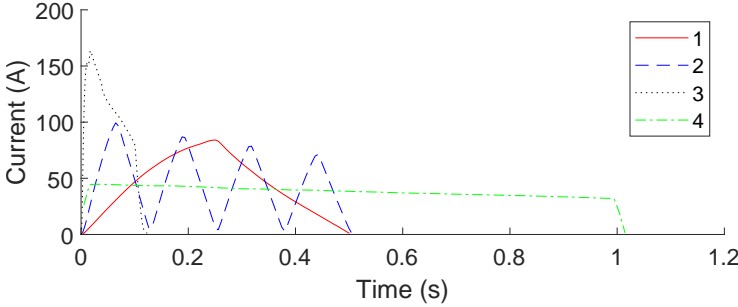

**Figure 12.** Current profiles for numerical thermal simulation study; Legend values correspond to tests described in Table 2.

The results of this study are shown in Figure 13. The maximum temperatures reached for each test are listed in Table 2. A critical analysis for this type of work was to determine the extent of any uncertainty that might exist in the results. For example, it was stated that a sample width of 0.635 mm was used in this testing. However, the actual sample range was between 0.6096 and 0.6604 mm. Temperature calculations rely on the measured resistivity of the sample, which had a range of 16.5–17.5 µΩ-cm. These deviations can be considered to determine the general possible deviation that exists for these tests. The following equation describes the trend of this deviation (the data used to fit this trend line can be found in the Appendix A as Figure A3):

$$\% \ uncertainty = \left(-4.915 \times 10^{-11}\right) \times T^{-5.088} + 0.2542 \tag{1}$$

where $T$ is the nominal temperature value in K. This equation fits the data with an $r^2$ value of 0.8815. For example, a test that results in a calculated temperature of 350 °C when using nominal dimensions and resistivity values may actually be between 306 and 394 °C.

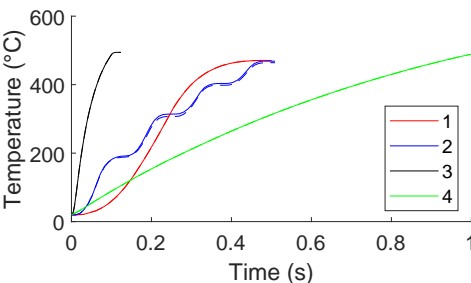

**Figure 13.** Temperature results of numerical simulation; Temperatures measured at the center of the geometry.

Figure 13 also shows the difference between the inclusion and exclusion of a constant temperature boundary condition applied at the grip location. Data that included this edge condition are shown as dashed lines; however, they are difficult to see since they are very similar to the solid-lined comparison. It can be concluded that this edge condition does not significantly influence the temperature at the center of the specimen, which is where these measurements were taken. The influence of this edge condition on the axial temperature profile of the specimen has already been discussed and can be seen in Figure 5. In summary,

with the 100 mm sample length that was utilized, this edge condition does not affect the maximum temperature of the sample.

Figure 13 also shows the difference in the temperature profiles of these tests. DC test conditions showed an approximately linear increase in temperature throughout the test, whereas pulsed tests showed undulations corresponding with their respective pulse frequencies. This results in minor changes in the sample's time-at-temperature. For example, although tests *1* and *2* had the same test length, test *1* can be seen as holding a higher temperature for a longer time than test *2*, whereas test *2* reached a higher temperature faster than test *1*. This difference will be considered in these analyses, as it is currently unknown if these minor differences can affect aging.

While the simulations performed in COMSOL to determine the temperature of samples can be solved relatively quickly ($\approx$10 s per test), temperature estimations were streamlined using MATLAB. Within this software, the temperature was estimated using temperature-dependent specific heat, thermal conductivity, and density values, which were also used in COMSOL. Convection cooling was also calculated. This resulted in relatively low differences between the MATLAB and COMSOL calculated temperatures. For the four tests previously discussed, the percent differences between these values were 1.3, 0.7, 0.2, and 0.2%.

### 2.4. Additional Pulse Testing

As was previously stated, the Ametek power supply was only capable of achieving a maximum of 16 Hz in a triangle waveform, which produced a much lower current ramp rate than was desired. Therefore, testing conditions were expanded by using a custom-built battery bank, along with custom inductors that were used to vary the ramp-rate of the battery bank output (more information on the inductors used for this testing can be found in a separate publication. The battery bank was controlled using a high-current MOSFET. An example test is shown in Figure 14.

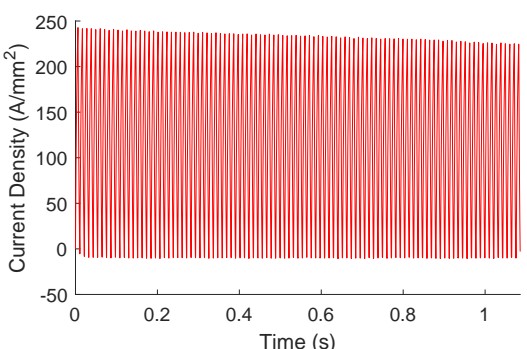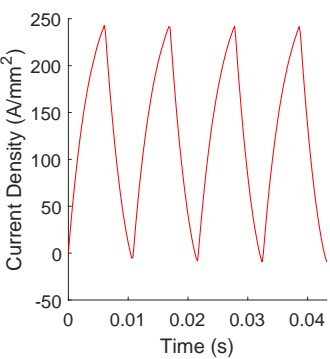

**Figure 14.** Example battery bank output: Added inductance: 1.672 mH, 36 V.

### 2.5. Ex Situ Resistivity Measurements

As previously stated, immediately following heat treatment and current applications, samples were placed into a $-30\,°C$ environment, where they remained until ex situ resistivity measurements were taken. These measurements were taken using either a Keithley 2001 or Keithley 2010 digital multimeter. The test leads were positioned 75 mm apart.

### 3. Results

Three repetitions of each test condition were performed. Data presented herein represent the average of these three tests and any error bars represent a single standard deviation.

The results of DC testing for all test conditions are shown in Figure 15. This establishes a baseline for this research work. An attempt was made to discern any effects of the time length of tests. This was accomplished by fitting trend lines to each test length and comparing the resulting plots. An example of this is shown in the Appendix A as Figure A4 (trend line fit was third order polynomial unless less than four data points were available.

Then second order polynomial was used). However, these trends were poorly fit and could not be used. An obvious difference between test lengths is also not observable in the plots shown in Figure 15. Therefore, it can be concluded that the time length of DC tests does not influence aging within the parameters explored herein. The trend lines shown in Figure 15 represent a fourth-order polynomial equation. Note that this trend line was forced to intercept the y-axis at the baseline -T6 resistivity value. The trend lines shown in this plot serve only to aid in the visualization of the data.

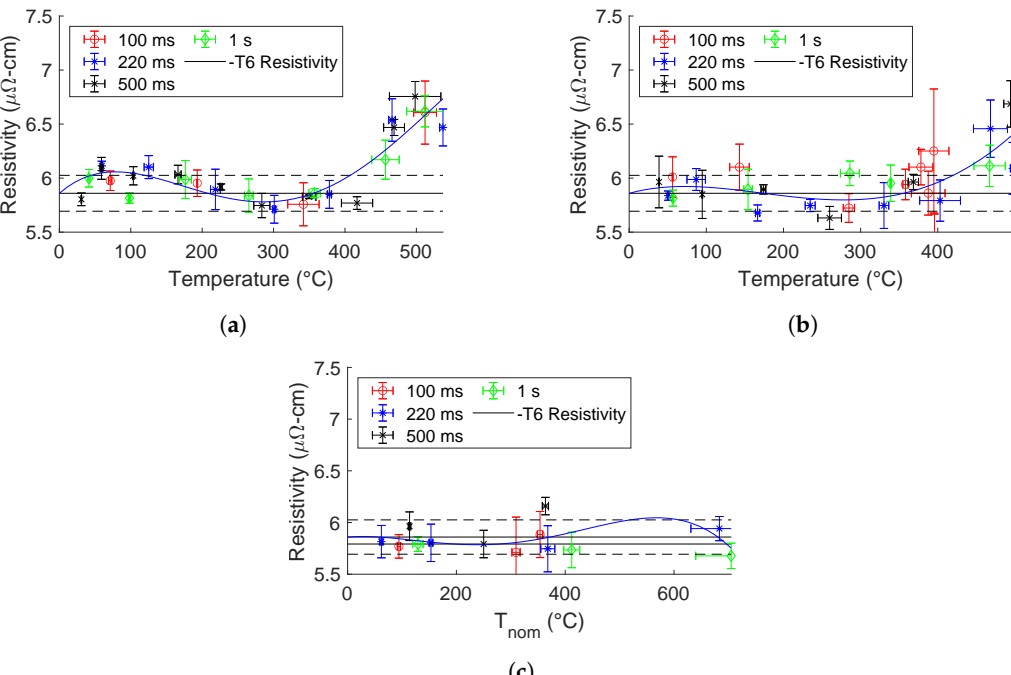

**Figure 15.** DC test results of ex situ resistivity measurements; Dashed black lines represent a single standard deviation of the -T6 baseline resistivity; (**a**) Air-cooled, air-quenched, (**b**) air-cooled, water-quenched, (**c**) water-cooled, water-quenched, temperature values are non-physical.

It was observed from this baseline DC testing that the water-cooled, water-quenched testing did not produce any significant trends deviating from the baseline resistivity value. Note that the pseudo maximum temperature of these tests was significantly greater than the air-cooled testing. This indicates that without cooling, these tests would have resulted in melting, indicating the amount of current which was passed through the sample during these tests. This observation provides good evidence that any effects of the electric current are simply the result of joule heating. Due to this initial conclusion, water-cooled, water-quench testing was not performed for pulsed testing.

The results of a downsampling of pulsed test data are shown in Figure 16 for 16 Hz frequency tests. The legend values in this figure represent the number of pulses that were applied. This can also be correlated with the relative length of the test time. For example, a 2-pulse test lasted for 0.125 s ($t_{period} = 1/16$ $t_{test} = 2 * t_{period}$). A significant similarity is observed for the air-cooled, air-quench testing trends. These trend lines are plotted together in Figure 17a, as well as the air-cooled, water-quenched trend lines in Figure 17b.

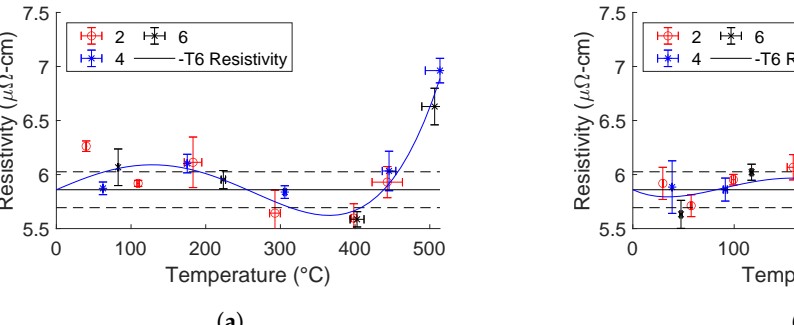

(**a**)　　　　　　　　　　　　　　　　(**b**)

**Figure 16.** 16 Hz test results of ex situ resistivity measurements, legend values indicate number of pulses applied; Dashed black lines represent a single standard deviation of the -T6 baseline resistivity; (**a**) Air-cooled, air-quenched, (**b**) air-cooled, water-quenched.

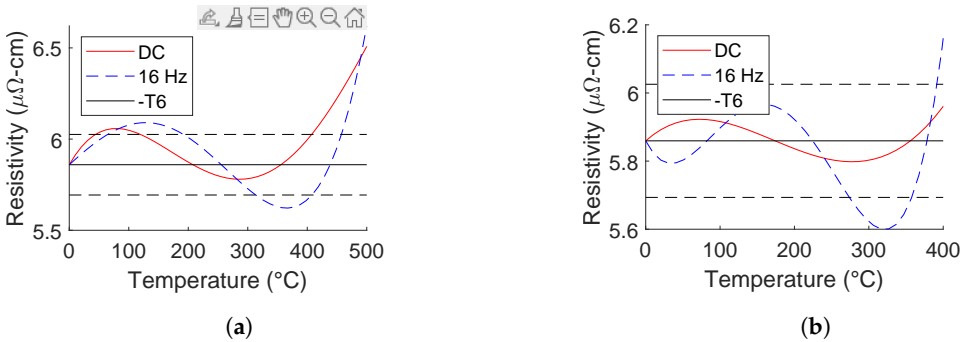

(**a**)　　　　　　　　　　　　　　　　(**b**)

**Figure 17.** Trend line comparison for DC and 16 Hz tests; (**a**) Air-cooled, air-quenched, (**b**) air-cooled, water-quenched.

Since no difference between tests of different time lengths was discovered, battery bank testing was only performed for 100 ms test lengths. A typical argument seen in EAM literature is that rapid pulsing at high amplitudes will produce an electroplastic effect. Performing tests under this condition maximizes the achievable current amplitude, testing this argument. The results of this testing are shown in Figure 18 and the parameters used are shown in Table 3. For each test condition, the battery bank was set to 36, 24, and 12 V to produce the three data points shown for each test ID. An example plot of the test condition which produced the highest current density with the fastest ramp rate is shown in Figure 19. The effects of changing the voltage of these tests are demonstrated in Figure A5 in the Appendix A. These test conditions are also plotted together in Figure A6 in the Appendix A.

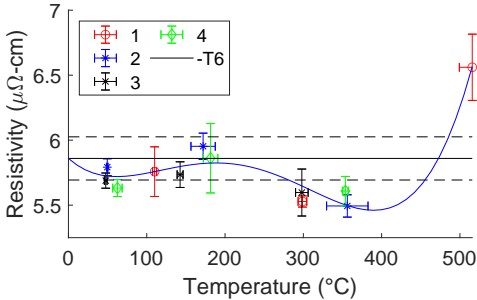

**Figure 18.** Results of testing using battery bank as power supply; Legend values correspond to parameters listed in Table 3.

**Table 3.** Battery bank parameters used in tests shown in Figure 18.

| Test ID | Added Inductance (μH) | Frequency (Hz) |
|---------|----------------------|----------------|
| 1 | none | 2.8 kHz |
| 2 | 206 | 310 Hz |
| 3 | 252 | 280 Hz |
| 4 | 414 | 140 Hz |

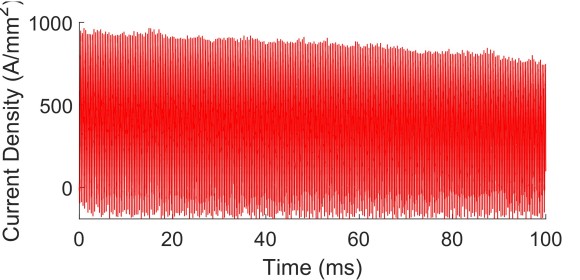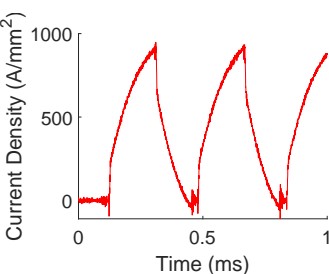

**Figure 19.** Current amplitude response of battery bank testing using Test ID 1 with 36 V.

## 4. Discussion

The trend lines of all air-cooled, air-quenched tests are shown in Figure 20a. This figure may show a difference between these trends. However, this is not apparent when considering the deviations from these tests. This is shown in Figure 20b. This plot is very effective in portraying a major conclusion of this work: Temperature is the most influential parameter of electrically assisted heat treatment of AA7075.

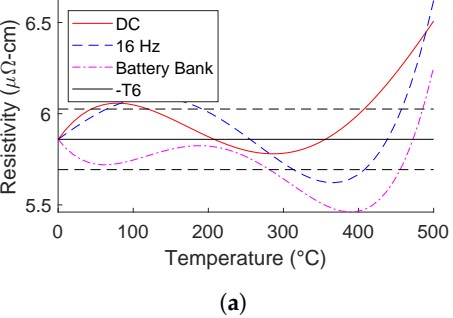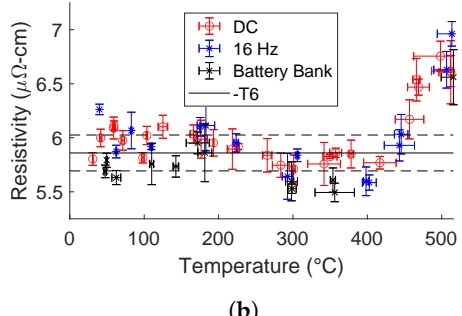

(**a**)                                                                 (**b**)

**Figure 20.** (**a**) Trend lines and (**b**) data points of all tests; Air-cooled, air-quenched.

This conclusion satisfies the primary objective of this dissertation work. However, it is beneficial to further distinguish between electrical and purely thermal effects. The testing performed herein is rather similar to retrogression heat treatment, where -T6 alloys are heated to temperatures below the solvus line for short lengths of time. Literature exists which shows the hardness results of various retrogression heat treatment cycles, shown in Figure 2. It is unclear from the article used to produce this plot [8] what the size of the specimens was and if quenching was performed after the retrogression heat treatment. Therefore, it is not possible to estimate an approximate time–temperature profile for these experiments with any certainty. As such, the data represented in this plot will be taken at face value (i.e., a data point showing a cycle of 240 °C for 20 s will be assumed to have a temperature profile which closely matches exactly that). In reality, there may be some discrepancy due to slow heating/cooling.

According to the thermal studies previously performed, the air-quenched test samples remain at relatively high temperatures for about 10 s (shown in Figure 8b). This can explain why there was no significant difference between tests of different time lengths. Considering a cooling time of 10 s, the heating time of these tests is insignificant, ranging between 0.1 and 10% of the cooling time in general. Therefore, for the analyses that follow, the test time length does not need to be considered.

An analysis can now be performed to see how well the trends observed in electrically assisted heat treatment align with conventional heat treatment retrogression. It is unclear from the literature what the effects of transient temperatures are, such as slow heating or cooling. As such, no literature exactly matches the cooling regime of this testing. A single temperature/time value was created by taking the average temperature across various time spans. The specimens' temperature during air cooling was interpolated from COMSOL simulation data, similar to the study performed to create the temperature profiles shown in Figure 8b. These data are shown in the Appendix A as Figure A7.

Figure 21 represents the results of this study. To better understand this plot, consider the following example. The red star data point presented at the five-second mark represents data from the present study that had an average temperature of $230 < T_{ave} < 250$ after five seconds of cooling, according to the simulation results shown in Figure A7. It can be stated generally then that the five-second analysis emphasizes that the maximum temperature the specimen reached is most influential on resistivity/hardness, whereas the 30-s analysis emphasizes that the time-at-temperature is most influential. Therefore, it was hypothesized that one of these analyses should align well with the hardness data from [8]. However, this was not necessarily observed. It is suggested that future testing attempts are made to match the temperature and holding times of these tests using resistive heating to perform a direct comparison.

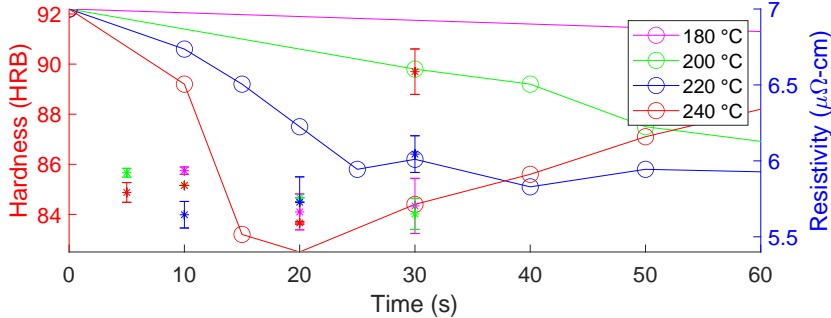

**Figure 21.** Comparison of electrically assisted heat treatment and hardness data presented in [8]: circle data points represent [8] hardness, star data points represent resistivity values from the present study.

The results of this study can also be compared to the furnace-heated experiments shown in Figure 7. These data showed a significant drop in resistivity after heating to 300 °C and an increase in resistivity after heating to 500 °C. A limitation of this study was the inability to perform rapid heating experiments, as previously mentioned. However, the results of this furnace-heated study align well with the electrically assisted results, which show a drop in resistivity around 300–400 °C, with an increase in resistivity at temperatures above 450 °C. It can therefore be concluded that no electroplastic effects are necessary for explaining the results discovered herein.

Figure 21 can also explain the difference between the air-cooled, air-quench, and air-cooled, water-quench experiments. It can be seen in this figure that there is little effect on hardness for tests that last less than 10 s. Since the water-quench tests were only at temperature for about one second or less, their resistivity was not significantly changed.

*Xu et al. Direct Comparison*

The study performed by Xu et al. [3] is described in greater detail in the Introduction section. In summary, this study found that electropulsing AA7075 resulted in nearly complete solutionization after a 220 ms pulse of AC current with an RMS current density of 200 A/mm$^2$. It is studied herein whether or not the use of AC or otherwise transient current is necessary to produce this result or if a similar result can be achieved with a nominally DC current. While this information can be effectively extrapolated from other data discussed herein, similar test parameters to Xu et al. [3] will be used for this section.

It was determined to use the Ametek power supply to test the comparable DC current. Ideally, the DC comparison test should have a rapid ramp rate to the desired current/current density (200 A/mm$^2$ in this case), should hold this current steadily throughout the test, then should quickly ramp down to zero after the desired amount of time. The maximum ramp-up/down rate was achieved using the battery bank without any inductors. This test is shown in Figure 22. This condition was produced by placing a 0.1 $\Omega$ resistor in the 12 V circuit without any inductor. The exact RMS current density for this test was 192 A/mm$^2$.

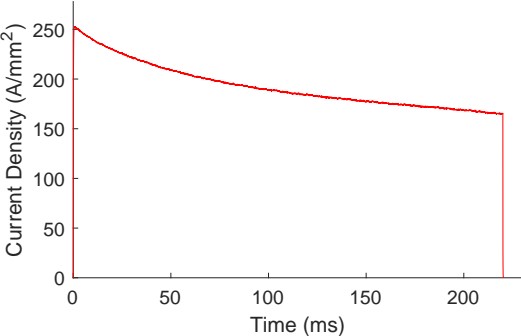

**Figure 22.** DC pulse of approximately 200 A/mm$^2$ using the battery bank power supply.

The comparable DC pulse using the Ametek power supply is shown in Figure 23, which had an exact RMS current density of 205 A/mm$^2$. The battery bank delivers a much more rapid ramp rate. However, this is at the cost of a rapidly declining current density. The majority of this decline is due to the batteries. A declining current density is also shown for the Ametek power supply. However, this is caused by the increased resistance of the sample resulting from its increasing temperature. An example of the water-cooled testing is shown in Figure 24. Since voltage control was used in this case, which offers the fastest ramp rate, the resulting current decreased as resistance increased, according to $I = V/R$. The ramp-up/down portion of this test was about 8.4%, whereas this value was about 0.3% for the battery bank. It was determined to use the Ametek for this testing, sacrificing the relatively slow ramp-up/down rate for a more consistent current density.

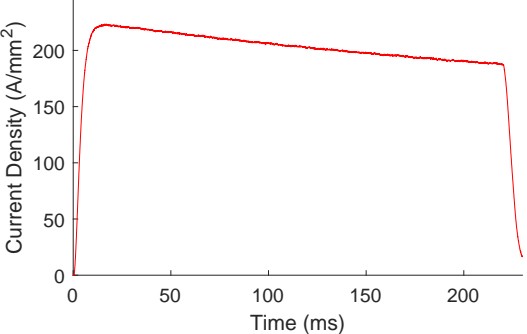

**Figure 23.** DC pulse of approximately 200 A/mm$^2$ using the Ametek power supply with voltage control; Air-cooled test condition.

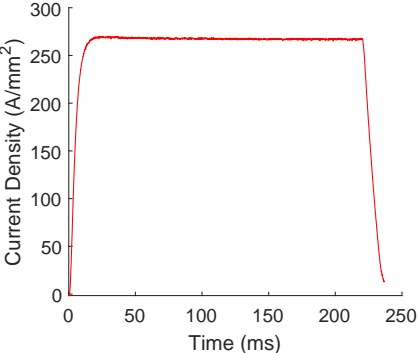

**Figure 24.** DC pulse of approximately 200 A/mm$^2$ using the Ametek power supply with voltage control; Water-cooled test condition.

The results of this study are compared to the test method of Xu et al. [3] in Figure 25. Surprisingly, even though there is no mention of any pre-testing performed in Xu et al.'s [3] work, the test parameters they used are closely aligned with the local minimum of resistivity discovered herein. It is unclear if this is purely a coincidence or if pretesting was omitted from their publication. Nevertheless, according to the previous conclusions cited herein, it is likely that no electroplastic effects are necessary to explain the results observed in Xu et al.'s [3] publication.

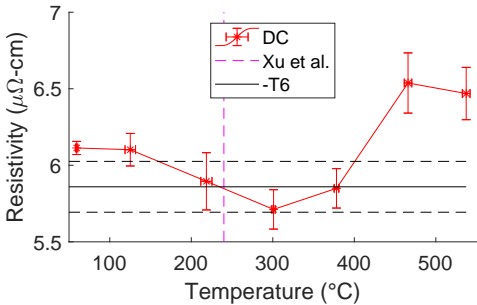

**Figure 25.** Direct comparison of testing described in Xu et al. [3] and DC pulse testing; Test length: 220 ms.

## 5. Conclusions

The results of this study have shown that resistive heating can be effectively used to perform rapid retrogression heat treatments of these alloys. There is no indication that an electroplastic phenomenon is needed to explain these results. Since the material's resistivity can be linked to precipitation density, which is also related to strength; when attempting to weaken this material, a maximum temperature of about 350 °C should be targeted. An especially relevant discovery of this testing is that a holding time at temperature was unnecessary. However, as was observed for water-quench testing, the material cannot be rapidly heated and immediately cooled. Instead, air-cooling provides sufficient time-at-temperature to affect change within the material. This suggests that the material can be rapidly heated and immediately air-cooled, significantly improving the efficiency of this process over conventional literature results. Additionally, these results indicate no difference between DC and pulsed/transient electrical conditions.

An example from the literature that made an electroplastic conclusion was challenged. It was determined that an electroplastic effect is not needed to explain the results of this testing; rather, the results can be explicitly attributed to joule heating.

**Author Contributions:** Conceptualization, T.G. and L.M.; methodology, T.G.; software, T.G.; validation, T.G.; formal analysis, T.G.; investigation, T.G.; resources, T.G.; data curation, T.G.; writing—original draft preparation, T.G.; writing—review and editing, T.G. and L.M.; visualization, T.G.; supervision, L.M.; project administration, L.M.; funding acquisition, L.M. All authors have read and agreed to the published version of the manuscript.

**Funding:** This research received no external funding.

**Data Availability Statement:** Data is available on request.

**Acknowledgments:** This material is based upon work supported by the National Science Foundation Graduate Research Fellowship Program under Grant No. 1744593. Any opinions, findings, and conclusions or recommendations expressed in this material are those of the authors and do not necessarily reflect the views of the National Science Foundation.

**Conflicts of Interest:** The authors declare no conflict of interest.

## Appendix A

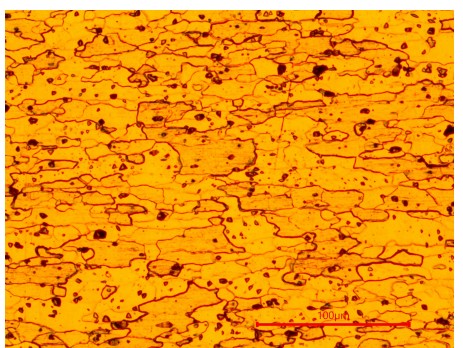 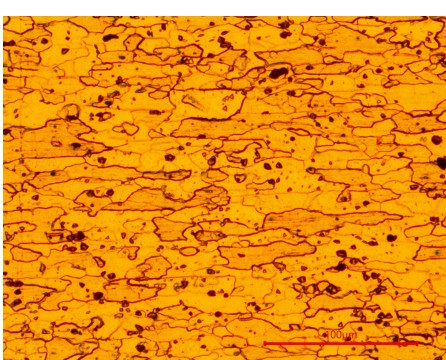

**Figure A1.** Example microstructure of two separate AA7075 sheets.

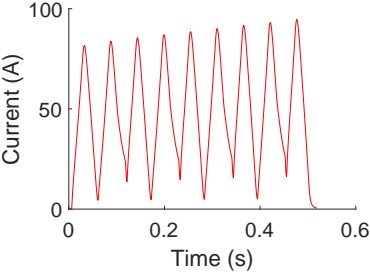

**Figure A2.** Ametek output at 18 Hz; Displays distorted waveshape.

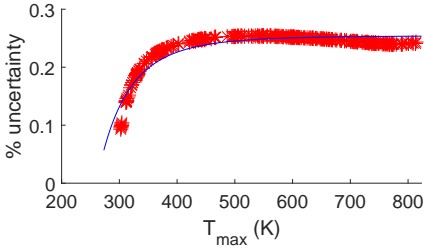

**Figure A3.** Trend of percent uncertainty for different maximum temperature values; red points indicate data, blue line indicates fitted trend.

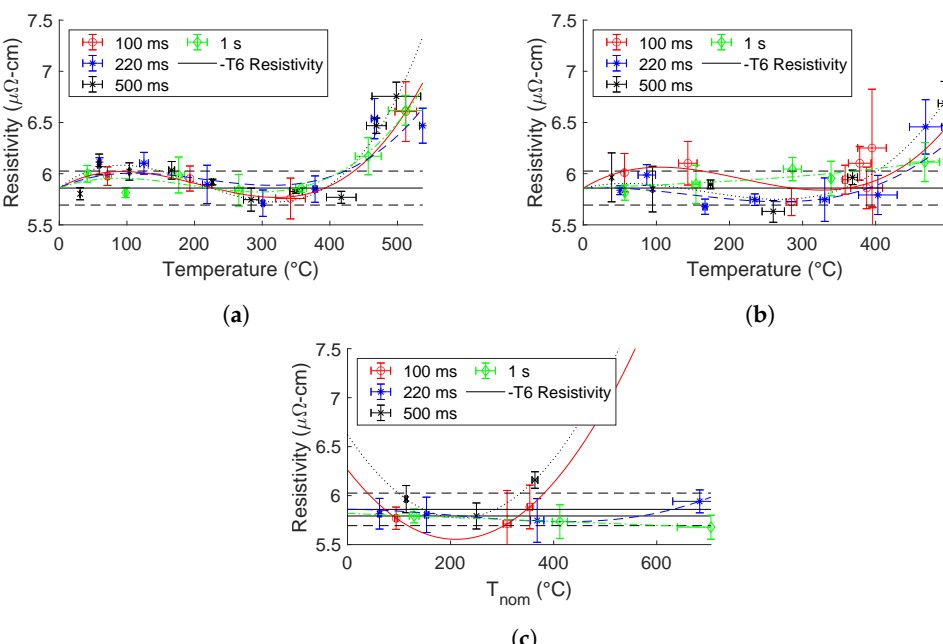

(**a**)

(**b**)

(**c**)

**Figure A4.** Attempted trend line fitting to DC test data; (**a**) Air-cooled, air-quenched, (**b**) air-cooled, water-quenched, (**c**) water-cooled, water-quenched, Temperature values are non-physical.

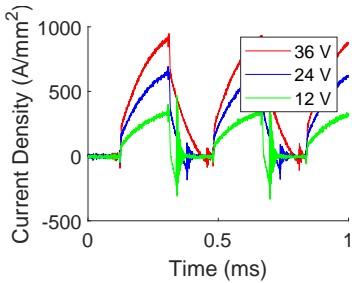

**Figure A5.** Example effect of changing voltage on power supply output.

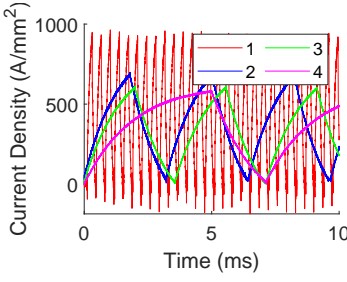

**Figure A6.** Output of power supply for all test IDs; 36 V.

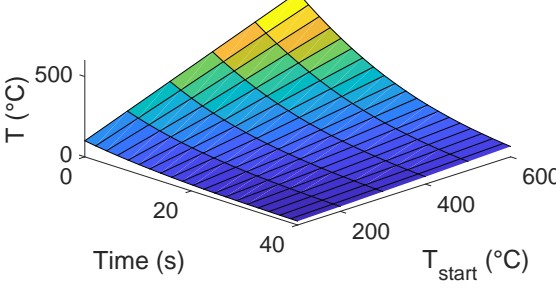

**Figure A7.** Specimen cooling at various starting temperatures.

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
