# Peer review of "Heat Treatment of AA7075 by Electropulsing and DC Current Application"

_jmmp, doi:10.3390/jmmp7020073_

Round 1
Reviewer 1 Report
The manuscript presents the study of the heat treatment of AA7075 under different conditions and the results were compared with the literature data. The authors conclude that the temperature is essential during the heat treatment of AA7075 and no difference of DC and pulsed/transient electrical conditions can be seen. The authors described their experiments in very detail and analyzed their results in a reasonable way. I would recommend the publication in the Journal after some minor corrections, e.g., some question marks (lines 21 and 327) and units’ errors (lines 147, 150, 175, 285, and in some tables).
Author Response
These issues have been properly addressed. Thank you for your time spent reviewing this research.
Reviewer 2 Report
The paper proposes using elecrtopulsing and DC current for heat treatment of AA7075. Various test parameters are explored to determine the difference between pulsing current and DC currents. The paper is well organized and readable. As a result, the paper can be accepted after some minor points can be revised as follows:
1. In page 1, 0. Introduction should be changed to 1. Introduction
2. In page 1, line 13 [1?] should be changed to [1]
3. Fig. 6(a), the label (a) is not clear.
4. The label Fig. 10 needs to change to Current waveform of Ametek Sorenson SGA 10/1200 power supply.
5. The quality of Fig. 21 can be improved.
6. The label (b) of Fig. A4 is not clear.
7. References [5], [6],, [7], [10], [12] are too old.
Author Response
- In page 1, 0. Introduction should be changed to 1. Introduction
There was an issue with the Latex template which has now been changed.
2. In page 1, line 13 [1?] should be changed to [1]
This issue has been corrected, as well as other formatting issues which may have shown "?" or omitted some characters.
3. Fig. 6(a), the label (a) is not clear.
The subfigure sizes have been adjusted to correct this.
4. The label Fig. 10 needs to change to Current waveform of Ametek Sorenson SGA 10/1200 power supply.
This caption has been revised.
5. The quality of Fig. 21 can be improved.
This figure has been redrawn at a larger scale.
6. The label (b) of Fig. A4 is not clear.
This was an issue with the Latex template and has been resolved.
7. References [5], [6], [7], [10], [12] are too old.
References 5, 6, and 7 were used as material property inputs to simulations. The age of such text is irrelevant since material properties will generally remain constant over time.
Reference 10 describes an electrically-assisted manufacturing test that investigated the electric current's effects on precipitate density. The three references that are provided related to this (9-11) are the only relevant publications. Therefore, since there are very few works to cite related to this, this particular citation will remain included.
Citation 12 is a heavily cited article that also pertains to material properties. Again, as such, the age of this text is irrelevant.
Reviewer 3 Report
n The summary of the article is not written systematically. And the content is incomplete and not highly generalized.
n Some of the innovation points in the introduction are not prominent enough and the summary is not good enough.
n References are cited so frequently and are generally outdated, such as reference 2.
n English language is poor.
n The description in the results section is not completely.
n The conclusion section should be divided into several parts and should be highly summarized.
Author Response
The summary of the article is not written systematically. And the content is incomplete and not highly generalized.
It is unclear what the reviewer is specifically referencing by this comment.
Some of the innovation points in the introduction are not prominent enough and the summary is not good enough.
There are very few innovation points related to this subject, as it is a fairly novel process. All of the innovation points have been summarized in the introduction.
References are cited so frequently and are generally outdated, such as reference 2.
This reference is less than 10 years old and should not be considered as outdated. There are relatively few publications related to electrically assisted manufacturing in general, with even fewer on the subject of electrically assisted heat treatment processes. This article, as well as many other articles cited herein, represent all of the publications related to the subjects they are referred to. As such, there does not exist any newer articles than those referenced herein.
English language is poor.
Some grammar/spelling corrections were made. It is unclear if the reviewer is criticizing any grammar/spelling errors or the general style of writing. While proficiency of writing style can be considered subjective to some extent, the authors do not agree with the reviewer's ranking of English quality. Therefore, major edits to the language used throughout will not be considered.
The description in the results section is not complete.
It is unclear what specifically is being criticized by this comment.
The conclusion section should be divided into several parts and should be highly summarized.
Additions have been made to the conclusion to address this comment.
Round 2
Reviewer 3 Report
I recognize the content of research work, but do not recognize the writing style of the article. The abstract of the article is too general and has not made any substantive modifications compared to the initial draft. Many details were not fully described, so this article must be rejected, if no substantive modifications are made.